# Conserving Marine Forests: Assessing the Effectiveness of a Marine Protected Area for *Cystoseira sensu lato* Populations in the Central Mediterranean Sea

**DOI:** 10.3390/plants13020162

**Published:** 2024-01-06

**Authors:** Francesco Paolo Mancuso, Gianluca Sarà, Anna Maria Mannino

**Affiliations:** 1Department of Earth and Marine Sciences (DiSTeM), University of Palermo, Viale delle Scienze Ed. 16, 90128 Palermo, Italy; gianluca.sara@unipa.it; 2NBFC—National Biodiversity Future Center, 90133 Palermo, Italy; annamaria.mannino@unipa.it; 3Department of Biological, Chemical and Pharmaceutical Sciences and Technologies, University of Palermo, 90123 Palermo, Italy

**Keywords:** macroalgal distribution, *Cystoseira sensu lato*, *Cystoseira*, *Ericaria*, *Gongolaria*, biodiversity conservation, Marine Protected Areas, Mediterranean Sea

## Abstract

Marine Protected Areas (MPAs) are vital for biodiversity conservation, yet their effectiveness in preserving foundation seaweeds remains understudied. This study investigates the diversity and distribution of *Cystoseira sensu lato* (including *Cystoseira*, *Ericaria,* and *Gongolaria*, hereafter referred to as *Cystoseira s.l.*) populations in an MPA located in the central Mediterranean Sea, comparing them with those in two unprotected sites. We hypothesized MPA *Cystoseira s.l.* populations would display higher diversity and structure compared to outside unprotected sites. Results revealed a total of 19 *Cystoseira s.l.* species at depths of 0–20 m, with the MPA exhibiting a higher diversity than unprotected sites. Thus, MPAs can play a crucial role in fostering the diversity of *Cystoseira s.l.* populations. However, no significant differences were observed among the MPA’s protection zones, raising questions about the zoning effectiveness. Additionally, our survey uncovered a substantial presence of non-indigenous seaweeds within the MPA. In conclusions, while MPAs improved *Cystoseira s.l.* diversity compared to unprotected sites, the varying efficacy of protection within MPA zones suggested a necessity for site-specific conservation strategies. The presence of non-indigenous seaweeds emphasizes ongoing challenges. This study provides a baseline for understanding *Cystoseira s.l.* population dynamics, crucial for future monitoring and conservation efforts in the face of global change.

## 1. Introduction

Marine forest seaweeds, mainly brown algae of the orders Fucales and Laminariales, form some of the most productive and diverse habitats on the world’s cold-water and temperate rocky reefs [1,2,3,4]. In the Mediterranean Sea, these foundation seaweeds mainly comprised species of the genus *Cystoseira* C. Agardh (Fucales, Phaeophyceae), recently divided into three genera: *Cystoseira*, *Gongolaria* Boehmer, and *Ericaria* Stackhouse [5,6] and hereafter referred to as *Cystoseira s.l.* (*sensu lato*). *Cystoseira s.l.* forests provide a wide range of ecosystem services, supporting a rich associated biota, including fish, invertebrates, and other algae [2,7,8]. They are essential foundation species, significantly enhancing the structural complexity and productivity of coastal communities from the surface down to the upper circalittoral zone [9,10,11,12]. Additionally, they are considered useful indicators of water and ecosystem quality according to the Water Framework Directive (2000/60/EC) and the Marine Strategy Framework Directive (2008/56/EC) [13,14].

Like other coastal marine ecosystems, these long-lived (between one and more than five decades [12,15]) primary producers are highly impacted, particularly close to urban areas, due to combined effects of anthropogenic impacts such as pollution, urbanization, the introduction of non-native species, overfishing, coastal aquaculture, and climate-change [2,16,17,18,19,20,21,22]. Consequently, the decline or loss of *Cystoseria s.l.* populations have been observed on many rocky coasts, leading to a shift from complex and productive benthic communities to less structured assemblages dominated by turf-forming algae, mussels, or sea urchin barrens [1,2,16,18,21,23,24,25,26]. This shift results in a decrease in essential ecosystem services [2,27].

Marine Protected Areas (MPAs) have become highly important tools for biodiversity conservation and management worldwide [28]. MPAs typically restrict human activities such as overfishing and urbanization, allowing natural regeneration and maintenance of marine ecosystems, fostering greater diversity [29]. In the Mediterranean Sea, a large part of the research concerning the effectiveness of MPAs has focused on fish assemblages [30], while few studies focused on foundation seaweeds [17,31,32]. MPAs can also play a critical role in the conservation of *Cystoseira s.l*. forests, as they guarantee protection from various anthropogenic impacts such as overfishing and urbanization [29,31]. The protective environment within MPAs may facilitate higher recruitment and reproduction rates of *Cystoseira s.l.* populations. Reduced habitat disturbance, in fact, can create conditions conducive to successful reproduction and the establishment of new individuals [29]. Moreover, the better protection provided by MPAs compared to the non-managed sites made them excellent areas where restoration actions of *Cystoseira s.l.* forests should be prioritized [29]. Healthy *Cystoseira s.l.* forests in MPAs may also represent an important source of propagules for the restoration of lost or degraded forests outside MPA borders, a crucial aspect due to the low dispersal capacity of most of these species [17,24,33,34].

In this study, we investigate the diversity and distribution of *Cystoseira s.l.* populations within the “Capo Gallo-Isola delle Femmine” MPA in the central Mediterranean Sea. This MPA was created in 2002 and affected the sea stretch between the towns of Palermo and Isola delle Femmine (Sicily, Italy), covering approximately 22 km^2^ of sea area and a coastline of about 16 km. The MPA is divided into three main zones, each with its own level of environmental protection (zone A: no-take zone, zone B: general protection, and zone C: partial protection) (Figure 1), with the aim of prohibiting harvesting, and in general, any activity that may constitute a danger or disturbance to vegetal and animal species, including the introduction of foreign species (D.M. 24-07-2002 Ministry for Environment, Land and Sea Protection) [35]. The MPA area is also identified as Site of Community Importance (SCI, ITA020047—Fondali di Isola delle Femmine Capo Gallo).

Moreover, we tested the effectiveness of the MPA for *Cystoseira s.l.* conservation by comparing these populations within the MPA to those in two unprotected sites (Figure 1). We hypothesized *Cystoseira s.l.* populations would be more diverse within the MPA compared to outside unprotected sites, and the areas within the MPA characterized by higher levels of protection would host highly diverse populations. Finally, in the investigated sites, we documented the presence of non-indigenous seaweeds, providing crucial information that offered a clearer understanding of the current challenges.

## 2. Results

A total of 19 *Cystoseira s.l.* species were found between 0 to 20 m of depth in the surveyed sites (Table 1 and Table 2). The highest number (seventeen species) was found within the “Capo Gallo-Isola delle Femmine” MPA, while Monte Cofano and Punta Barcarello hosted nine and five species, respectively (Table 1). 

### 2.1. Cystoseira s.l. Assemblage within the MPA

Within the MPA, ten *Cystoseira s.l.* species were found into zone A, while fourteen were found in zone B and C (Table 1), with six species (*C. compressa*, *C. humilis* var. *myriophilloides*, *E. amentacea*, *E. brachycarpa*, *E. crinita*, and *G. montagnei* var. *compressa*) consistently found across different levels of protection. 

Whitin the MPA, zone A exhibited lower abundances (average cover percentage) and a lower number of *Cystoseira s.l.* species compared to the other two levels of protection (B and C) (Figure 2a,b). The values for these two metrics differed between zone B and C, with zone B showing slightly lower average abundance but higher number of species compared to zone C (Figure 2a,b). Shannon diversity and Pielou’s evenness were higher in zone A and B compared to zone C (Figure 2c,d).

The *Cystoseira s.l.* assemblage differed significantly among the different MPA zones (PERMANOVA results model: F_2,17_ = 55.027 and 40.394 for structure and composition, respectively, *p* < 0.001; Appendix A), explaining a substantial amount of the total variation (R2 = 70.73% and 64.27% for structure and composition, respectively). Additionally, there was a significant difference among sites within MPA zones (PERMANOVA results model: F_3,17_ = 10.706 and 10.711 for structure and composition, respectively, *p* < 0.001; Appendix A), accounting for a moderate amount of the total variation (R2 = 20.64% and 25.56% for structure and composition, respectively; Appendix A). No significant difference was found for transects within sites and MPA zones (Appendix A). The Principal Coordinate Analysis (PCoA) ordination plot clearly separated the structure of the *Cystoseira s.l.* assemblage among MPA zones, with zone A at a far distance from zone B, while sites in zone C placed among the other two protection zones, with Punta Barcarello being closer to sites in zone B and Punta Matese to zone A (Figure 3a). *Cystoseira s.l*. composition, instead, clearly separated zone A from B and C, with partial overlap of the *Cystoseira s.l.* composition of the site of La Cala (within zone B) with sites of zone B (Figure 3b). 

### 2.2. Comparison of Cystoseira s.l. Assemblage between MPA and Unprotected Sites

When comparing the MPA with unprotected sites, urban and white sites exhibited higher average of *Cystoseira s.l.* abundance, comparable to zones C and B within the MPA rather than zone A (Figure 4a). The species richness of the white and urban sites was comparable to that within the MPA, with the white site being closer to zone C, while the urban site showed lower values more comparable to those of zone A (Figure 4b). Shannon diversity and Pielou’s evenness of the white site were comparable to those of zone C, while those of the urban site showed lower values, although no significant differences were detected (Figure 4c,d).

When comparing the *Cystoseira s.l.* assemblage between the MPA zones and unprotected sites (Appendix A), it was revealed there were significant differences at the urban site compared to both the MPA’s and white sites (Figure 5a,b), with an average dissimilarity of 71% and 60% for structure and composition, respectively (Appendix A). On the other hand, the *Cystoseira s.l.* assemblage of the white site was more similar to the MPA’s sites (Figure 5a,b) with an average similarity of 55% and 59% for structure and composition, respectively. It was closer to the Punta Matese site (zone C) with an average similarity of 64% and 67% for structure and composition, respectively (Appendix A).

Here, we provided an overall description of *Cystoseira s.l.* populations across the investigated sites. Of the nineteen discovered species, *C. humilis* showed the largest depth adaptability (range depth = 0–20 m) followed by four species (*C. foeniculacea* f. *tenuiramosa*, *G. montagnei*, *G. montagnei* var. *tenuior* and *G. sauvageauana*) found between 5 and 20 m depth (Table 2). At all investigated sites, *E. amentacea* grew on the outer margin of vermetid where it formed dense and continuous belts across the MPA’s sites and the Monte Cofano (white) site; however, this species was not found at the Punta Priola (urban) site. Sparse thalli of *C. compressa* f. *rosetta* were also found at the sites of Isola and Punta Barcarello, while *C. compressa* was present within both the MPA and the two unprotected sites. *Cystoseira compressa* formed either small dense patches with thalli of about 10 cm high (without aerocystis) or individual thalli up to 40 cm high with branches provided by aerocystis. Between 1 and 5 m depth, *E. brachycarpa* formed large and extended forests within the MPA at the sites of Punta Barcarello, La Cala, and Isola, whereas at Punta Matese, Barcarello, and Capo Gallo, despite being dense, the populations assumed a discontinuous pattern due to the presence of large rocky boulders placed on rocky carbonate platforms. Dense and continuous populations *of E. brachycarpa* were also found at Monte Cofano, whereas at Punta Priola *E. brachycarpa* stands were patchy and less dense, intercalated by thalli of *Dictyopteris polypodioides* (De Candolle) J. V. Lamouroux. *Ericaria crinita* was the second most common species discovered between 1 and 5 m depth. At the sites of Barcarello, Punta Barcarello, and Capo Gallo, this species formed dense patches covering big boulders or, as in the other sites, its thalli were widely spread on the substrates and surrounded by other macroalgae. *Cystoseira humilis* was found at the site of La Cala and Monte Cofano as individual sparse thalli or dense patches. Finally, only at the site of Monte Cofano sparse thalli of *E. mediterranea* were found.

Between 5 and 10 m depth, *G. montagnei* var. *tenuior* was the most abundant which was found within the MPA at the sites of La Cala, Punta Matese, Punta Barcarello, Barcarello, and Monte Cofano. This species forms extensive dense forests, which can be seen in particular in La Cala and Punta Barcarello. In the other sites, *G. montagnei* var. *tenuior* grew in small patches or as individual thalli surrounded by other macroalgae. *Cystoseira humilis* var. *myriophylloides* on the other hand, was only found within the MPA at Isola, La Cala, Punta Matese, and Barcarello as isolated or groups of few thalli surrounded by other seaweeds, whereas *C. foeniculacea* was found only at the sites of Barcarello and Punta Priola.

The larger part of *Cystoseira s.l.* species (seven species) was found between 10 and 20 m depth (Table 2). *Ericaria funkii* was only found within the MPA at the sites of La Cala, Punta Matese, Punta Barcarello, and Barcarello. This species has a green iridescence and was found as groups of two to three individuals so close to be perceived as a single big individual. *Cystoseira foeniculacea* f. *latiramosa* and *C. foeniculacea* f. *tenuiramosa* were found as scattered individual thalli within the MPA, the first one at the sites of La Cala, Punta Matese, and Barcarello, and the second at La Cala, Punta Matese, Punta Barcarello, and Barcarello. *Ericaria dubia* was discovered in small patches within the MPA (Isola, La Cala, and Punta Barcarello) in areas with high sedimentation, with flattened primary branches of light brown color emerging from the sediment. Finally, *E. brachycarpa* var. *claudiae* was found as isolated individual thalli within the MPA (Punta Barcarello, Barcarello, and Capo Gallo) and at the unprotected site of Punta Priola (Table 2).

### 2.3. Non-Indigenous Seaweeds 

During data collection within both the MPA and unprotected sites, the presence of non-indigenous seaweeds was observed, regardless of the level of protection. In particular, four species were discovered: *Asparagopsis taxiformis* (Delile) Trevisan, *Caulerpa cylindracea* Sonder, *Caulerpa taxifolia* var. *distichophylla* (Sonder) Verlaque, Huisman & Procaccini, and *Lophocladia trichoclados* (C. Agardh) F. Schmitz (Figure 6). Thalli of *A. taxiformis* (Figure 6A) can be found across the entire “Capo Gallo-Isola delle Femmine” MPA, regardless of the protection zone and depth range. The species was discovered from 1 m to 20 m depth with a particularly high density in the Isola delle Femmine site. Moreover, *A. taxiformis* was found as epiphyte on *Cystoseira s.l.* species, especially *G. montagnei* var. *tenuior*. *Caulerpa cylindracea* (Figure 6B) was also observed across the MPA and did not appear to have a preferred depth. It can be found from the mediolittoral zone (intertidal rocky pools, vermetid reef cuvettes) down to the 20 m. While stolons can attain enormous densities, the vertical frond of the alga is not always apparent. *Caulerpa taxifolia* var. *distichophylla* (Figure 6C) was found between 0 and 10 m at Punta Barcarello and Barcarello, particularly on rocky substrate covered with sediment. Finally, blooms of *L. trichoclados* (Figure 6D) were detected over the MPA during the summer, capable of completely covering vast areas of substrate and all the seaweeds inhabiting them. This species has also been found to epiphyte *G. montagnei* var. *tenuior*. 

## 3. Discussion

According to our surveys, the MPA has more diverse *Cystoseira s.l.* populations in terms of number of species (Figure 7) than outside unprotected sites (Figure 8), confirming the hypothesis that MPAs can be an effective strategy for preserving and restoring these important foundation seaweeds [17,31,32]. Although this observation remains true when comparing *Cystoseira s.l.* populations across the MPA and external unprotected sites, our findings demonstrate no variations among the various degrees of protection within the MPA. Therefore, our initial hypothesis posited the effects of protection within the MPA would lead to significant differences in the diversity and abundance of these communities between areas with total protection and those with less protection, and was thus unfounded.

Studies have shown some Mediterranean MPAs were unable to protect or restore Fucalean algal forests, implying some of them were merely “paper parks” where regulations were not enforced [32,37]. We believed the lack of efficacy in protection observed within the MPA was due to the fact that the different zones within the MPA were most likely designed to protect fish stocks rather than Fucalean algal forests, despite the primary aims of the MPA stating otherwise when it was created (D. 24-07-2002 Ministry for Environment, Land and Sea Protection). Moreover, the observed differences between the zones could be explained by variations in seabed geomorphic features across MPA sites. In fact, even though the investigated sites presented the same exposure (northwest winds), local seabed conformation could be critical to shape seaweeds assemblage. Our observations suggested when the seabed consists of a gently sloping rocky carbonate platform, as seen at the MPA sites of La Cala and Punta Barcarello, we found more continuous and dense *Cystoseira s.l.* populations compared to other sites, such as Capo Gallo and Barcarello, where the presence of large scattered rocky boulders (Riggio and Raimondo, 1991 [38]; Lucido et al., 1992 [39]) created a discontinuous environment with heterogeneous light conditions, where small patches and individual thalli of *Cystoseira s.l.* were most common.

The distribution of macroalgae is intricately tied to the geomorphological characteristics of the seabed, a factor that plays a pivotal role in shaping coastal marine ecosystems. Geomorphological features such as substrate type, topography, and hydrodynamic conditions significantly influence the establishment and composition of macroalgal communities [40]. Substrate characteristics, for instance, directly impact algal attachment and growth, with different species displaying preferences for specific substrates [41,42]. Furthermore, hydrodynamic conditions, influenced by factors like wave exposure and water flow, contribute to the transport of reproductive propagules and nutrients, influencing the distribution and diversity of macroalgal communities [43]. While our study may not have incorporated specific geomorphological data, we recognized the importance of these characteristics and their potential role in explaining observed differences in macroalgal distribution among protection zones. Therefore, studies that will take into account seabed geomorphology will further clarify its role in the distribution of *Cystoseira s.l.* populations.

Data on *Cystoseira s.l.* species distribution within the “Capo Gallo-Isola delle Femmine” MPA are scarce and mainly date back to at least thirty years, making them only partially helpful for comparison. Giaccone and Sortino (1964) [44] reported the presence of *C. compressa*, *E. mediterranea*, *E. crinita,* and *G. barbata* on the seabed of Isola. Of these, *E. mediterranea* and *G. barbata* were not found in our surveys. In 1985, data from *G. barbata* e *G. montagnei* were reported at Capo Gallo and Isola [45], while more recent data reported *E. amentacea*, *Ericaria crinita,* and *G. montagnei* at the sites of Punta Barcarello and Capo Gallo [46,47]. The scarcity of historical data on *Cystoseira s.l.* species distribution within the MPA emphasizes the significance of our study as a baseline for understanding how these populations change in the future.

The comparison with unprotected sites revealed differences that were more evident when comparing the MPA sites with the Punta Priola site than with the site of Monte Cofano. The vegetation in Punta Priola was mainly characterized by high sedimentation rates, with *Dictyopteris polypodioides* being the main macroalgal species covering large part of the rocky substrate, which appeared to inhibit the growth of other seaweeds (Figure 8B). This pattern aligns with findings from previous studies, which emphasize the role of sedimentation in shaping not only *Cystoseira s.l.* populations [48] but macroalgal communities in general. Sedimentation is shown to inhibit zoospore adhesion [49] and facilitate negative interactions by promoting turf-forming algae, which, in turn, inhibit canopy-forming macrophytes [21,50,51]. The absence of *E. amentacea*, one of the most important *Cystoseira s.l.* species used to measure water quality, further revealed the site’s impacted state [52]. Water quality has been shown to reduce the survival and growth of *Cystoseira s.l.* [53], which would then justify the low diversity of these habitat-forming seaweeds observed in Punta Priola. Furthermore, the proximity of this site to urban center can facilitate the presence of multiple co-occurring anthropogenic stressors likely drivers of the poor *Cystoseira s.l.* conditions of this site [20]. In contrast, our findings demonstrate *Cystoseira s.l.* populations are comparable between the unprotected site of Monte Cofano and the sites within the “Capo Gallo-Isola delle Femmine” MPA (Figure 8A). The presence of healthy and dense *Cystoseira s.l.* forests in this unprotected site suggests anthropogenic disturbances, such as trampling, harvesting, pollution, and overgrazing, are relatively limited at Monte Cofano. This finding confirms robust forests thrive in non-protected, naturally isolated, and lightly disturbed locations [32].

In addition to the above observations, it is important to mention the presence of four non-indigenous seaweeds (*A. taxiformis*, *C. cylindracea*, *C. taxifolia* var. *distichophylla*, and *L. trichoclados*) within the MPA and at the unprotected sites. Non-indigenous species are one of the major threats to the Mediterranean Sea [54,55]. *Asparagopsis taxiformis* has been named one of the top 100 invasive seaweeds in the Mediterranean Sea [55]. The presence of *A. taxiformis* need further investigation because it has negative effects on *Cystoseira s.l.* populations, eroding biomass of primary producers and the associated biodiversity [56]. In accordance with other research [57,58,59], *C. cylindracea* was largely observed within the MPA area across different habitats and types of substrate, regardless of the levels of protection. It was found across all the investigated depth range, but thalli were also found at depths of 35 m (Mancuso’s personal observations). This suggests although MPAs are a useful management tool for the protection of biodiversity, they are still vulnerable to non-indigenous seaweeds [59,60]. The presence of *C. cylindracea* can have a negative impact on native seaweeds assemblages, also facilitating the subsequent invasion of a trophic specialist that takes advantage of niche opportunities that are created by the algae [61,62,63]. Finally, *L. trichoclados* could affect the structure of macrofauna associated with habitat forming seaweeds of *Cystoseira s.l.* [60] or cause the mortality of seagrasses [64].

## 4. Materials and Methods

The study was performed on the shallow rocky substrate (0 to 20 m depth) within the “Capo Gallo-Isola delle Femmine” MPA (Lat: 38.213961, Long: 13.277121) and two unprotected sites, Monte Cofano (Lat: 38.114429, Long: 12.677827) and Punta Priola (Lat: 38.192074, Long: 13.358161), located in the northwestern coast of Sicily, Italy (Figure 1), which were not subjected to marine protection.

### 4.1. The “Capo Gallo-Isola delle Femmine” MPA

The “Capo Gallo-Isola delle Femmine” MPA, established in 2002 by the Italian Ministry of Environment and Protection of Land and Sea, affected the sea stretch between the towns of Palermo and Isola delle Femmine. Covering approximately 22 km^2^ of sea area and a coastline of about 16 km, it is bounded to the east by the gulf of Mondello and to the west by the bay of Carini. An imposing calcareous dolomitic mountain crest (Capo Gallo, 562 m a.s.l.) defines the coastal strip, resulting in a steep and rocky coastal morphology. Due to the limestone nature, flowing waters generate karst phenomena, leading to caves of significant ecological importance (Grotta dell’Olio and Grotta della Mazzara). Only towards the western part, the rocky coast assumed a flat conformation, enlivened by the presence, about 300 m from the mainland, of the Isola delle Femmine (also known as Isola di Fuori), an isolated vestige of the aforementioned calcareous ridge. 

The MPA is divided into three main zones (A, B, and C), each with its own level of environmental protection (Figure 1). There are two no-take/no-access zones (zone A, total area of 1 km^2^, Figure 1), one in the north sector of Isola delle Femmine and the other in the stretch of sea at the west of Capo Gallo promontory, between the Puntazza and the Capo Gallo lighthouse. Zones B and C are buffer zones where human use restrictions, including fishing, become progressively lower. In particular, there are three general protection zones (zone B, total area of 2 km^2^, Figure 1), while the remaining sea within the MPA’s border includes a partial protection area (zone C, total area of 19 km^2^, Figure 1). The MPA area is also identified as Site of Community Importance (SCI, ITA020047—Fondali di Isola delle Femmine Capo Gallo).

### 4.2. Unprotected Sites

To analyze the effectiveness of MPA protection, we chose two unprotected marine regions: one natural and less impacted site named Monte Cofano and one highly impacted site close to the urban center called Punta Priola (Figure 1). The Monte Cofano site is located in front of the coast of the natural terrestrial reserve of the Monte Cofano promontory, near Custonaci and San Vito Lo Capo. Although it is not a marine reserve, its status as a terrestrial reserve offers some level of protection by restricting access from the land. Furthermore, there are no large urban areas nearby, and pollution is virtually absent. However, there are no restrictions on marine activities (like fishing or harvesting) in the area. The Monte Cofano site can then be considered an unaltered unprotected site based on its characteristics (white site). The site of Punta Priola was located between Mondello and Palermo. This site is clearly influenced by several anthropogenic stressors (urban site). The shoreline is densely developed, with small untreated outfalls (particularly in summer), and the little Rousvelt harbor located approximately 300 m west of the site. Moreover, the site attracts bathers who pour onto the seashore, resulting in an increase in stressors from trampling and harvesting activities.

### 4.3. Survey of Cystoseira s.l. Populations 

Scuba diving surveys were conducted at 6 sites within the “Capo Gallo-Isola delle Femmine” MPA, representing a large part of the MPA and the 3 levels of protection (2 sites for each level of protection), while surveys were carried out at 1 site in the uncontrolled sites. The sites were mostly exposed to northwest winds and had a similar seabed environment with carbonate platforms and rocky substrates. At each site, 3 belt transects [65] from 0 to 20 m depth were used to determine the distribution of *Cystoseira s.l.* species. Transect length changed according to seabed degradation, while width was 6 m (3 m left and right the transect). For each bathymetric range (0–5 m, 5–10 m, 10–15 m, and 15–20 m), cover percentage of *Cystoseira s.l*. species were estimated in 4 quadrats (50 × 50 cm) haphazardly selected (Appendix A). Cover was estimated dividing the quadrat into 25 equal squares: we attributed a cover score from 0 to 4 to each square, and then summed up scores where the taxon was present. Organisms filling <1⁄4 square were given the value of 0.5 [66]. During each dive, the water visibility was at least 10 m, allowing easy identification of *Cystoseira s.l.* thalli. The depth range was chosen to allow safe scuba diving (diving constraints such as decompression schedules and air consumption normally limited depths to less than 20 m) and to give access to the majority of *Cystoseira s.l.* species. 

All surveys were carried out in May, when the thalli of *Cystoseira s.l.* species in this area reached their maximum development [67]. Furthermore, pictures of the landscape were acquired to describe the status of the *Cystoseira s.l.* populations. Collection of thalli was limited to species that were difficult to identify in the field. Sampled thalli were deposited in the algological laboratory (Department STEBICEF—University of Palermo).

### 4.4. Data Analysis

For each investigated depth range, the abundance (N, average percentage cover), frequency (F%, the percentage of samples in which a particular species was present), and dominance index (D%, cover percentage of a particular species to the total cover percentage of *Cystoseira s.l.* species within the sample) for each *Cystoseira s.l*. species were estimated [68]. Additionally, for each area investigated (MPA’s zones and unprotected sites), *Cystoseira s.l.* species were characterized based on abundance (N; expressed as cover percentage), rarefied species richness (S), Shannon–Wiener diversity index (H′), and Pielou’s evenness index (J). 

Analyses of variance (ANOVAs) were used to test: (i) differences in the *Cystoseira s.l.* indices (N, S, H′, J) among the three protection zones (fixed factor with 3 levels: zone A, zone B, and zone C) within the MPA; (ii) differences among zones within the MPA and external unprotected sites (fixed factor with 5 levels: zone A, zone B, zone C, white, and urban). Besides the main factor in each analysis, ANOVAs included the factors site (random factor nested within zone) and transect (random factor nested within site and zone). 

Louvain community detection [69] was performed to detect the *Cystoseira s.l.* set for each considered depth range. Differences in the *Cystoseira s.l.* structure (which took into account species identity and relative abundance) and composition (presence/absence, which only took into account species identity) among sites were assessed by Permutational Multivariate Analysis of Variance (PERMANOVA). The analyses were based on a Bray–Curtis distance matrix of square-root transformed cover percentage of *Cystoseira s.l.* using 9999 permutations [70]. A principal coordinate analysis (PCoA) plot was generated to visualize the variation in *Cystoseira s.l.* assemblage structure (based on a Bray–Curtis distance matrix) and composition (based on Jaccard distance matrix). 

Statistical analyses were carried out in R open access statistical software version 4.1.2 [71].

## 5. Conclusions

Our findings emphasize the importance of the “Capo Gallo-Isola delle Femmine” MPA as a valuable tool for Fucalean forests of the genera *Cystoseira*, *Ericaria,* and *Gongolaria*, as well as a good reference for monitoring the temporal evolution of these foundation seaweeds. Further research should be conducted to gain more insights into *Cystoseria s.l.* species present in the MPA, particularly at depths greater than 20 m. Up until now, data on deeper *Cystoseira s.l.* species within the “Capo Gallo-Isola delle Femmine” MPA came from point-like observations. In particular, we have observed the presence of *Ericaria zosteroides* (C. Agardh) Molinari & Guiry and *G. montagnei* var. *compressa* were at a depth of 35 to 40 m at the site of Isola and the presence of *G. barbata* (Stackhouse) Kuntze at a depth of around 1 m in the port of Isola delle Femmine. This information remarks the MPA’s role in preserving diverse *Cystoseira s.l.* populations. It also emphasizes the importance of doing extensive habitat mapping of these key foundation seaweeds to monitor their range and health status. This is crucial in understanding how these valuable foundation species respond to the effects of global change.

## Figures and Tables

**Figure 1 plants-13-00162-f001:**
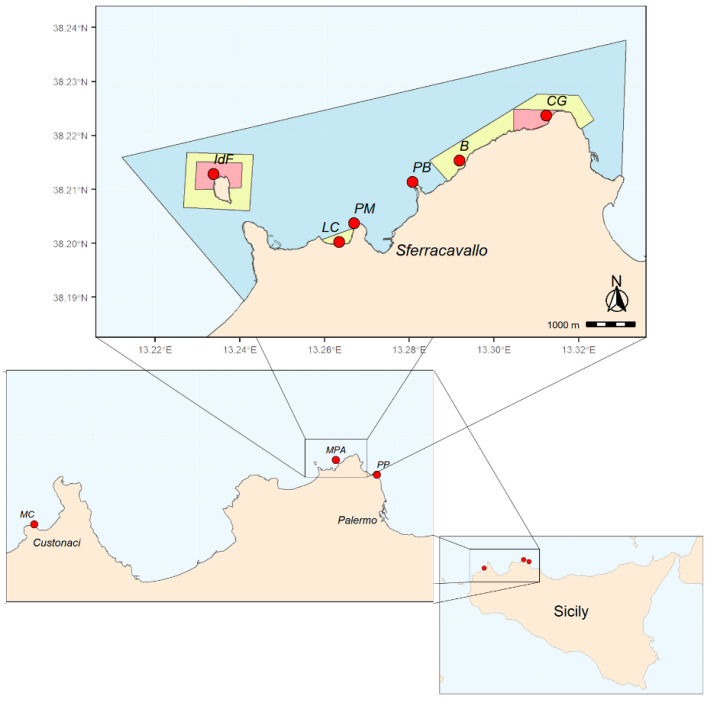
Study sites (red dots) along the northwestern rocky-shore of Sicily: *MPA* = MPA of “Capo Gallo-Isola delle Femmine”, *MC* = Monte Cofano, *PP* = Punta Priola. The top panel displays information about the sites investigated within the MPA: *IdF* = Isola, *LC* = La Cala, *PM* = Punta Matese, *PB* = Punta Barcarello, *B* = Barcarello, *CG* = Capo Gallo. Within the MPA (upper panel), red, yellow, and blue areas denote no-take (zone A), general (zone B), and limited (zone C) protection zones, respectively.

**Figure 2 plants-13-00162-f002:**
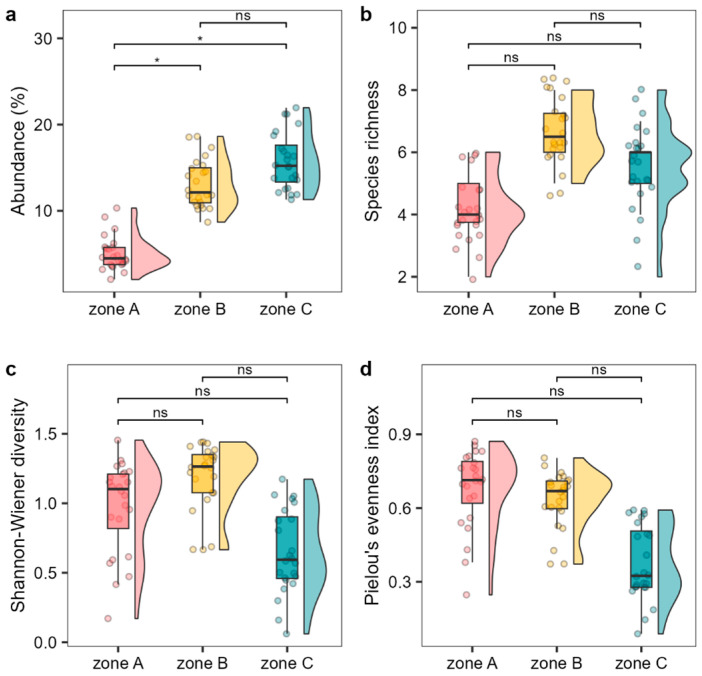
Abundance (**a**, average cover percentage), species richness (**b**), Shannon–Wiener diversity (**c**), and Pielou’s evenness index, (**d**) of the *Cystoseira s.l.* assemblage among the different MPA protection levels. Boxplots show extreme and lower whisker (vertical black line), lower and upper quartile (box), and median (horizontal black line). Density plot is shown beside each boxplot. Dots are raw data (n = 24). Significance codes: * *p* < 0.01, ns: *p* > 0.05.

**Figure 3 plants-13-00162-f003:**
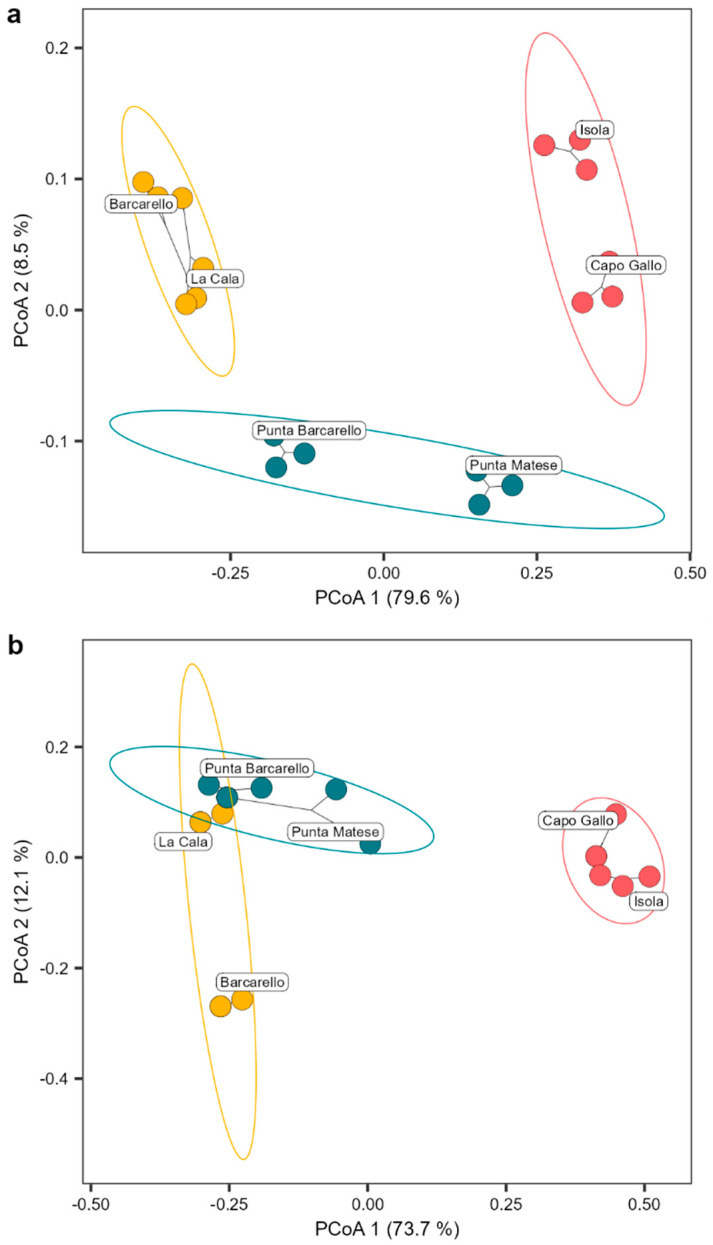
Structure (**a**) and composition (**b**) of the *Cystoseira s.l.* assemblage found. Circles show the 95% confidence of interval for each MPA zone (red = zone A, yellow = zone B, and blue = zone C). Principal coordinate analysis plot (PCoA) based on Bray–Curtis measure of square-root transformed *Cystoseira s.l.* percentage cover (structure) or Jaccard measure (composition).

**Figure 4 plants-13-00162-f004:**
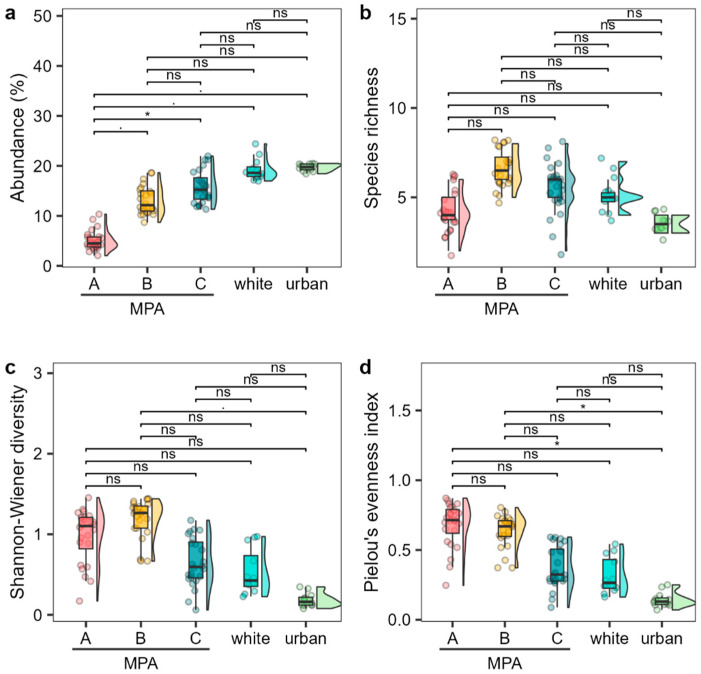
Abundance (**a**, average cover percentage), species richness (**b**), Shannon–Wiener diversity (**c**), and Pielou’s evenness index (**d**) of the *Cystoseira s.l.* assemblage among the different MPA zones and outside unprotected sites. Boxplots show extreme and lower whisker (vertical black line), lower and upper quartile (box), and median (horizontal black line). Density plot is shown beside each boxplot. Dots are raw data (n = 12–24). Significance codes: * *p* < 0.01, . *p* < 0.05, ns: *p* > 0.05.

**Figure 5 plants-13-00162-f005:**
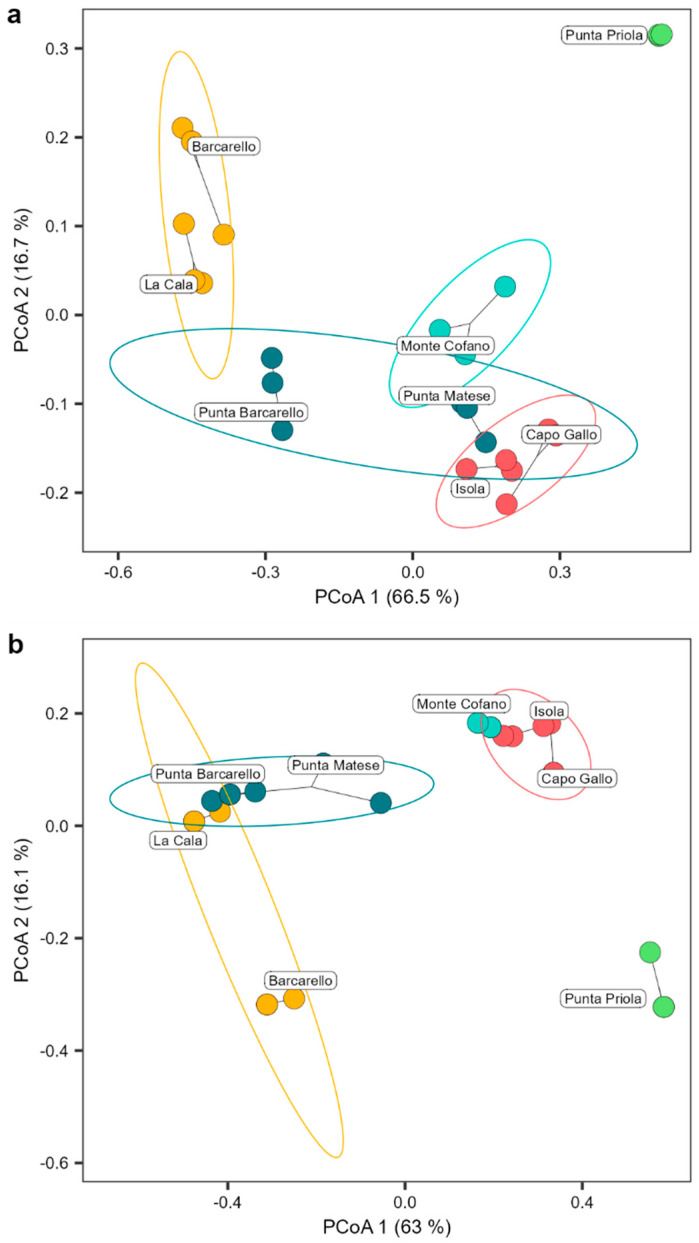
Structure (**a**) and composition (**b**) of MPA and outside unprotected sites *Cystoseira s.l.* assemblage. Circles show the 95% confidence of interval for each MPA zone (red = zone A, yellow = zone B, blue = zone C, light blue = white site and green = urban site). Principal coordinate analysis plot (PCoA) based on Bray–Curtis measure of square-root transformed *Cystoseira s.l.* percentage cover (structure) or Jaccard measure (composition).

**Figure 6 plants-13-00162-f006:**
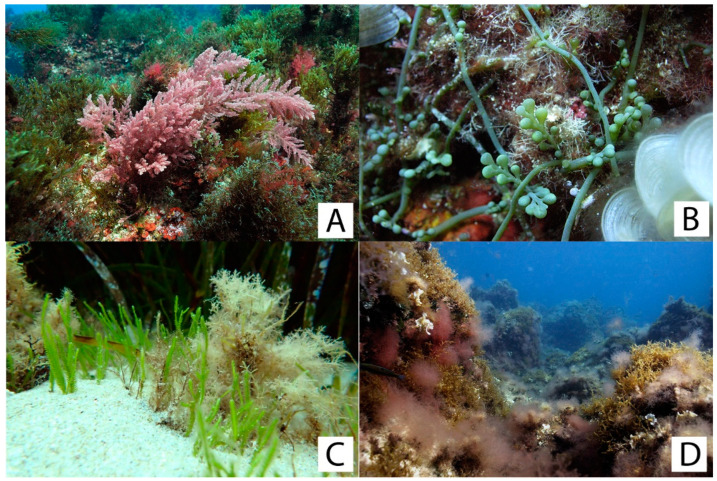
Invasive species found within the MPA of Capo Gallo-Isola delle Femmine: (**A**) = *Asparagopsis taxiformis*; (**B**) = *Caulerpa cylindracea*; (**C**) = *Caulerpa taxifolia* var. *distichophylla*; and (**D**) *= Lophocladia trichoclados*. Photos by Francesco Paolo Mancuso.

**Figure 7 plants-13-00162-f007:**
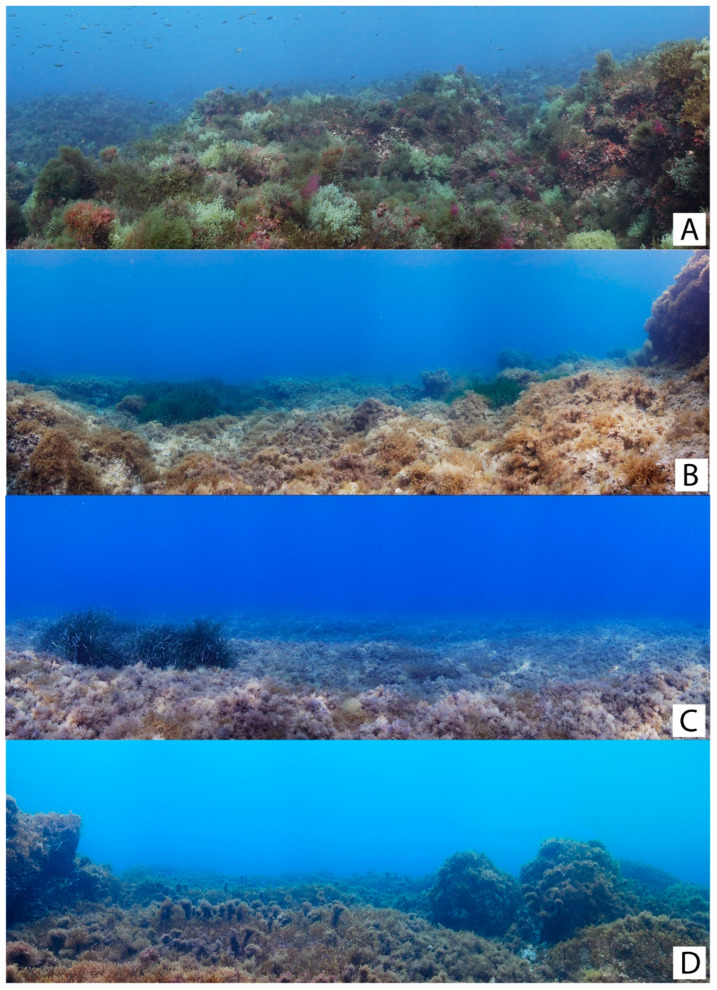
Examples of benthic habitats with *Cystoseira s.l.* populations within the MPA of “Capo Gallo-Isola delle Femmine”: (**A**) = Isola (zone A); (**B**) = Barcarello (zone B); (**C**) = La Cala (zone B); and (**D**) = Punta Barcarello (zone D). Photos by Francesco Paolo Mancuso.

**Figure 8 plants-13-00162-f008:**
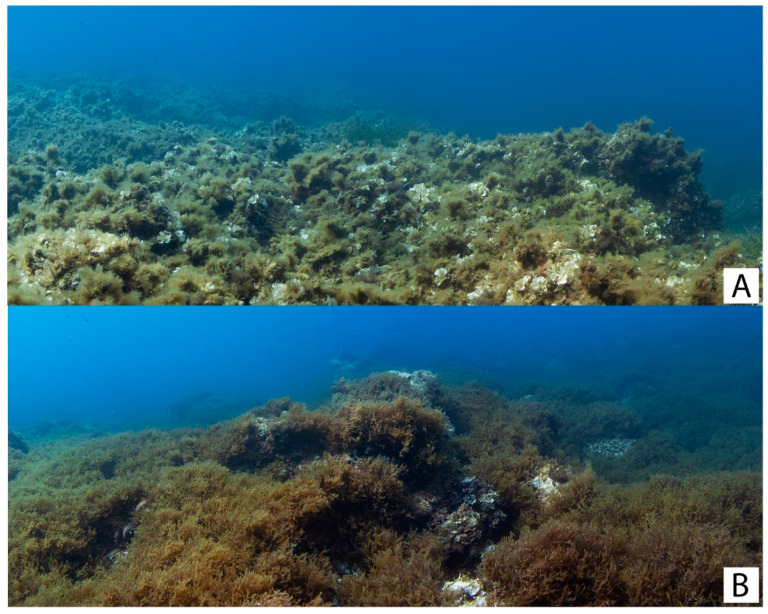
Examples of benthic habitats at the two unprotected sites: (**A**) = Monte Cofano characterized by the presence of well stated *Cystoseira s.l.* populations; (**B**) = Punta Priola, dominated by *Dictyopteris polypodioides.* Photos by Francesco Paolo Mancuso.

**Table 1 plants-13-00162-t001:** List of the recorded *Cystoseira s.l.* species across MPA zones (A, B, and C) and unprotected sites (white = natural and less impacted site, urban = highly impacted site close to the urban center). Taxonomy and nomenclature were updated according to the AlgaeBase [36] database. The plus symbol (+) indicates the presence of a species.

		MPA Zones			Unprotected Sites	
		A	B	C	White	Urban
Species						
*Cystoseira compressa*	(Esper) Gerloff & Nizamuddin 1975	+	+	+	+	+
*Cystoseira compressa* f. *rosetta*	(Ercegovic) Cormaci, G. Furnari, Giaccone, B. Scammacca & Serio	+		+		
*Cystoseira foeniculacea*	(Linnaeus) Greville		+			+
*Cystoseira foeniculacea* f. *latiramosa*	(Ercegovic) A. Gómez Garreta, M.C. Barceló, M.A. Ribera & J.R. Lluch 2001		+	+		
*Cystoseira foeniculacea* f. *tenuiramosa*	(Ercegovic) A. Gómez Garreta, M.C. Barceló, M.A. Ribera & J. Rull Lluch		+	+		
*Cystoseira humilis*	Schousboe ex Kützing		+	+	+	
*Cystoseira humilis* var. *myriophylloides*	(Sauvageau) J.H. Price & D.M. John	+	+	+		
*Ericaria amentacea*	(C. Agardh) Molinari & Guiry	+	+	+	+	
*Ericaria brachycarpa*	(J. Agardh) Molinari & Guiry	+	+	+	+	+
*Ericaria brachycarpa* var. *claudiae*	Boudouresque, Perret-Boudouresque & Blanfuné	+	+			+
*Ericaria crinita*	(Duby) Molinari & Guiry	+	+	+		
*Ericaria dubia*	(Valiante) Neiva & Serrão 2022		+	+		
*Ericaria funkii*	(Schiffner ex Gerloff & Nizamuddin) Molinari & Guiry		+	+		
*Ericaria mediterranea*	(Sauvageau) Molinari & Guiry				+	
*Gongolaria montagnei*	(J. Agardh) Kuntze	+		+	+	+
*Gongolaria montagnei* var. *compressa*	(Ercegovic) Verlaque, Blanfuné, Boudouresque & Thibaut	+	+	+	+	
*Gongolaria montagnei* var. *tenuior*	(Ercegovic) Molinari & Guiry		+	+	+	
*Gongolaria sauvageauana*	(Hamel) Molinari & Guiry				+	
*Gongolaria squarrosa*	(De Notaris) Kuntze	+				

**Table 2 plants-13-00162-t002:** Average abundance (N), frequency of occurrence (F %), and dominance (D %) of the *Cystoseira s.l*. species identified in MPA zones (A, B, and C) and unprotected sites (white = natural and less impacted site, urban = highly impacted site close to the urban center). Calculations are based on *Cystoseira s.l.* cover percentage using 50 × 50 cm quadrats (n = 4).

			MPA Zones									Unprotected Sites					
			A			B			C			White			Urban		
Depth Range	Species		N	F%	D%	N	F%	D%	N	F%	D%	N	F%	D%	N	F%	D%
0–5 m	*Cystoseira compressa*		3.5	41.7	3.8	8.3	50.0	4.7	5.4	37.5	5.1	7.9	50.0	6.7	11.7	50.0	51.9
	*Cystoseira compressa* f. *rosetta*		1.0	12.5	1.1				0.4	4.2	0.4						
	*Cystoseira humilis*					15.2	54.2	8.6	4.4	25.0	4.1	13.3	50.0	11.3			
	*Ericaria amentacea*		50.8	100.0	54.5	82.3	100.0	46.3	58.1	100.0	54.8	60.0	100.0	50.7			
	*Ericaria brachycarpa*		37.9	95.8	40.6	71.9	100.0	40.4	37.7	87.5	35.6	31.7	83.3	26.8	10.8	50.0	48.1
	*Ericaria mediterranea*											5.4	25.0	4.6			
5–10 m	*Cystoseira humilis* var. *myriophylloides*		3.1	12.5	11.0	8.3	20.8	8.0	4.4	16.7	6.3						
	*Cystoseira humilis*					9.0	37.5	8.6	6.0	29.2	8.7	3.3	25.0	7.3			
	*Cystoseira foeniculacea* f. *tenuiramosa*					8.1	37.5	7.8	7.5	29.2	10.8						
	*Cystoseira foeniculacea*					3.3	16.7	3.2							5.0	33.3	54.5
	*Ericaria crinita*		19.2	70.8	67.6	34.8	79.2	33.5	23.5	58.3	34.1						
	*Gongolaria montagnei*		6.0	33.3	21.3				3.1	16.7	4.5	6.7	16.7	14.7	4.2	33.3	45.5
	*Gongolaria montagnei* var. *tenuior*					40.2	83.3	38.8	24.6	62.5	35.6	25.0	75.0	55.0			
	*Gongolaria sauvageauana*											10.4	41.7	22.9			
10–15 m	*Cystoseira foeniculacea* f. *tenuiramosa*					11.2	50.0	12.3	5.2	25.0	14.5						
	*Cystoseira humilis*					10.6	37.5	11.6									
	*Cystoseira foeniculacea*					5.6	25.0	6.2							2.1	8.3	26.3
	*Ericaria crinita*		12.7	45.8	43.9	14.2	45.8	15.5	6.7	33.3	18.6						
	*Ericaria brachycarpa* var. *claudiae*		1.2	8.3	4.3	6.0	16.7	6.6							4.2	16.7	52.6
	*Ericaria funkii*					14.0	45.8	15.3	6.7	25.0	18.6						
	*Gongolaria montagnei* var. *compressa*		5.4	25.0	18.7	4.0	12.5	4.3	1.9	12.5	5.2	0.8	8.3	3.2			
	*Gongolaria montagnei* var. *tenuior*					25.8	58.3	28.2	11.2	50.0	31.4	10.8	25.0	41.9			
15–20 m	*Cystoseira foeniculacea* f. *latiramosa*					7.9	37.5	11.9	2.7	20.8	10.2						
	*Cystoseira foeniculacea* f. *tenuiramosa*				7.7		33.3	11.6	1.0	4.2	3.9						
	*Cystoseira humilis*				3.5	16.7	5.3										
	*Ericaria dubia*				4.6	25.0	6.9	1.7	12.5	6.2							
	*Ericaria funkii*					14.8	37.5	22.3	4.2	16.7	15.6						
	*Gongolaria montagnei*		15.6	41.7	41.0				10.0	41.7	37.5	8.3	33.3	52.6	5.4	33.3	100.0
	*Gongolaria montagnei* var. *compressa*		14.8	70.8	38.8	5.0	16.7	7.5	3.5	20.8	13.3	4.2	33.3	26.3			
	*Gongolaria squarrosa*	7.7		29.2	20.2												
	*Gongolaria montagnei* var. *tenuior*					22.9	54.2	34.5	3.5	12.5	13.3						
	*Gongolaria sauvageauana*										3.3		16.7	21.1			

## Data Availability

Data are contained within the article and Appendix A.

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
