# Peer review of "Conserving Marine Forests: Assessing the Effectiveness of a Marine Protected Area for Cystoseira sensu lato Populations in the Central Mediterranean Sea"

_plants, 2024, doi:10.3390/plants13020162_

Round 1
Reviewer 1 Report
Comments and Suggestions for Authors
As has been shown by numerous studies, the mediterranean decline of macroalgal forests is raising major concerns for the potentially negative consequences on biodiversity and ecosystem functions. Protecting and restoring these habitats requires detailed information on their distribution, ecological status, and drivers of decline. Also has been stated that it is urgent to prioritize Cystoseira s.l. forests as one of the main objectives of the protection and management of the Mediterranean marine environment. Legal protection measures would be critical for implementing conservation and restoration strategies for these canopy-forming macroalgae. The present work is a valuable contribution in that direction.
I include some comments and corrections:
Ln. 62-68.- In the last paragraph of the introduction, it is necessary to briefly advance some of the information that is subsequently provided in material and methods related to the MPA (lines 355-375) (date of creation, surface, biological characteristics that supported its creation, a definition of zones A, B and C, etc.) with the purpose of having essential information to interpret the data presented in the results. It is evident that the MPA was not created to protect the Fucales forests, but other organisms (Posidonia, fish), so this is an important matter to highlight.
Since there is not enough information on the situation of the Fucales forests in the area prior to the declaration of the MPA, it is risky to draw conclusions regarding the benefits of protection for the Fucales forests, as might be expected. However, the information provided on the current situation of these forests is very useful as a basis for subsequent evaluations.
Table 1.- I suggest arranging the species in alphabetical order (genus, species, form) in the table.
Table 2. Ln. 93: replace (Fr %) with (F %). Delete the authorships column of the taxa that were already included in table 1. The data in the table for each species can be compared more easily.
Ln. 180 and others: There is no uniformity in abbreviating the generic name when naming the species in the text. Sometimes the full name is written and other times it appears abbreviated, even though they are species from three different genera that are named simultaneously.
Ln. 226: delete 1845
Ln. 227, 237, … The authors should consider that recently Golo et al (2023) have shown that Lophocladia lallemandii is a synonym of Lophocladia trichoclados. [Golo et al 2023 (https://doi.org/10.1080/09670262.2023.2260443)]
Ln. 277-285.- The geomorphological characteristics of the bottoms seem to be important in explaining differences between the different sampling sites. This aspect has not been considered in the analyses and, however, is considered very relevant to explain some differences observed between Fucales forests. The authors should analyse and discuss something more about this matter (protection zone vs. bottom geomorphology).
Author Response
Dear reviewer,
We appreciate the feedback and comments you provided, which have enhanced the quality of the manuscript. Our responses follow the reviewer's comments.
Sincerely
Francesco Paolo Mancuso
Reviewer #1:
As has been shown by numerous studies, the mediterranean decline of macroalgal forests is raising major concerns for the potentially negative consequences on biodiversity and ecosystem functions. Protecting and restoring these habitats requires detailed information on their distribution, ecological status, and drivers of decline. Also has been stated that it is urgent to prioritize Cystoseira s.l. forests as one of the main objectives of the protection and management of the Mediterranean marine environment. Legal protection measures would be critical for implementing conservation and restoration strategies for these canopy-forming macroalgae. The present work is a valuable contribution in that direction.
I include some comments and corrections:
Comment - Ln. 62-68.- In the last paragraph of the introduction, it is necessary to briefly advance some of the information that is subsequently provided in material and methods related to the MPA (lines 355-375) (date of creation, surface, biological characteristics that supported its creation, a definition of zones A, B and C, etc.) with the purpose of having essential information to interpret the data presented in the results. It is evident that the MPA was not created to protect the Fucales forests, but other organisms (Posidonia, fish), so this is an important matter to highlight.
Response –Thank you for your suggestion. We have revised the paragraph to include brief information about the Marine Protected Area (MPA). While we acknowledge that the MPA was likely established to protect fish and Posidonia rather than Fucales species, we currently lack a specific reference supporting this claim. The official law that created the MPA outlines one of its main aims as “to prohibit harvesting, and in general, any activity that may constitute a danger or disturbance to vegetal and animal species, including the introduction of foreign species.” This information has been incorporated into the modified paragraph: “In this study, we investigate the diversity and distribution of Cystoseira s.l. populations within the “Capo Gallo - Isola delle Femmine” MPA in the central Mediterranean Sea. This MPA was created in 2002 and affects the sea stretch between the towns of Palermo and Isola delle Femmine (Sicily, Italy), covering approximately 22 km2 of sea area and a coastline of about 16 km. The MPA is divided into three main zones each with its own level of environmental protection (zone A: no-take zone, zone B: general protection, and zone C: partial protection) (Figure 1), with the aim to prohibit harvesting, and in general, any activity that may constitute a danger or disturbance to vegetal and animal species, including the introduction of foreign species (D. 24-07-2002 Ministry for Environment, Land and Sea Protection). The MPA area is also identified as Site of Community Importance (SCI, ITA020047 – Fondali di Isola delle Femmine Capo Gallo).”
Comment - Since there is not enough information on the situation of the Fucales forests in the area prior to the declaration of the MPA, it is risky to draw conclusions regarding the benefits of protection for the Fucales forests, as might be expected. However, the information provided on the current situation of these forests is very useful as a basis for subsequent evaluations.
Response – Thank you for your observation. As we stated in the conclusion our study establishes a baseline for understanding the population dynamics of Cystoseira s.l., which is crucial for future monitoring and conservation efforts in the face of climate change.
Comment - Table 1.- I suggest arranging the species in alphabetical order (genus, species, form) in the table.
Response – Done.
Comment - Table 2. Ln. 93: replace (Fr %) with (F %). Delete the authorships column of the taxa that were already included in table 1. The data in the table for each species can be compared more easily.
Response – Done.
Comment - Ln. 180 and others: There is no uniformity in abbreviating the generic name when naming the species in the text. Sometimes the full name is written and other times it appears abbreviated, even though they are species from three different genera that are named simultaneously.
Response – Thank you for the observation. We modified abbreviation.
Comment - Ln. 226: delete 1845
Response – Done.
Comment - Ln. 227, 237, … The authors should consider that recently Golo et al (2023) have shown that Lophocladia lallemandii is a synonym of Lophocladia trichoclados. [Golo et al 2023 (https://doi.org/10.1080/09670262.2023.2260443)]
Response – Thank you for your suggestion. We updated the name of Lophocladia lallemandii with L. trichoclados (C.Agardh) F.Schmitz.
Comment - Ln. 277-285.- The geomorphological characteristics of the bottoms seem to be important in explaining differences between the different sampling sites. This aspect has not been considered in the analyses and, however, is considered very relevant to explain some differences observed between Fucales forests. The authors should analyse and discuss something more about this matter (protection zone vs. bottom geomorphology).
Response – While we do not possess specific geomorphological data in this study, we agree that acknowledging the significance of these features is crucial. In our revised manuscript, we included a dedicated section discussing the importance of geomorphological characteristics in macroalgal distribution, supported by relevant scientific references. We believe this addition will enrich the discussion and provide a more comprehensive understanding of the factors influencing the observed variations among protection zones. “The distribution of macroalgae is intricately tied to the geomorphological characteristics of the seabed, a factor that plays a pivotal role in shaping coastal marine ecosystems. Geomorphological features such as substrate type, topography, and hydrodynamic conditions significantly influence the establishment and composition of macroalgal communities [1]. Substrate characteristics, for instance, directly impact algal attachment and growth, with different species displaying preferences for specific substrates [2,3]. Furthermore, hydrodynamic conditions, influenced by factors like wave exposure and water flow, contribute to the transport of reproductive propagules and nutrients, influencing the distribution and diversity of macroalgal communities [4]. While our study may not have incorporated specific geomorphological data, we recognize the importance of these characteristics and their potential role in explaining observed differences in macroalgal distribution among protection zones. Then, studies that will take into account seabed geomorphology will further clarify its role in the distribution of Cystoseira s.l. populations.”

Reviewer 2 Report
Comments and Suggestions for Authors
This work provides a comprehensive qualitative and quantitative description of the diversity and abundance of Cystoseira fucoids in a specific region of Sicily, Italy. It examines locations with varying anthropogenic pressure, including different areas within a Marine Protected Area (MPA) with a variable protection regime. It also includes a "White site" outside the reserve with low alteration and one highly impacted site close to an urban area.
From my perspective, this study lacks replicated locations under comparable conditions, making it inappropriate to designate them as "control sites." Consequently, the results can only be generalized with stronger bibliographic support in the discussion. The observed trends also need more detailed discussion and thorough comparison with existing literature.
In the current global change scenario, the qualitative and quantitative information provided in this study is relevant and will enable future research on compositional changes (qualitative and quantitative) in these foundational Mediterranean seaweeds. Overall, I believe this work is publishable but needs restructuring based on the assessments provided.
Below, I explain the aspects that I think can be improved in each section:
Abstract: It should be restructured with the new contributions from the introduction and discussion. While the abstract emphasizes the role of climate change in driving Cystoseira communities, in the introduction authors recognize various anthropogenic stressors such as overfishing, urbanization, and pollution. Therefore, it might be more appropriate to refer to "global change" rather than "climate change."
Introduction:
The hypothesis that Cystoseira s.l. will be more diverse and structured within the MPA and most protected situations is proposed, but the causal mechanisms supporting this hypothesis need clarification, along with adequate bibliographic support. The concept of structure should be defined in the introduction as it is an ambiguous concept. On the other hand, the economic value of Cystoseira communities is discussed to emphasize their importance for conservation, citing a study (25) that draws an analogy between the energy investment required for ecosystem maintenance and a transformation of this investment into monetary units. Despite acknowledging the relevance of the cited study, I do not believe it can be stated in the introduction that these communities have direct economic value, as they are not commodities directly entering the market system and, therefore, cannot possess such value. I think the aspect of monetizing natural communities (which presents many implementation problems, see, for example, Vatn 2000: https://doi.org/10.3197/096327100129342173) is not the focus of this study. I would omit the mention of "economic value" in line 51.
Materials and Methods:
There are 3 zones (ABC) depending on the degree of protection within the MPA. However, in Figure 1 these zones (A, B, C) are not marked. It would be easier to understand if this information were also in Figure 1 (although these zones are well described in the text). On the other hand, more detail on the type of protection regime present in zones B and C is lacking.
The unprotected locations are contrasted from the perspective of anthropogenic impact. It must be said, therefore, that this study does not have replicated localities under comparable conditions, nor can we speak, in my opinion, of "control sites" since there is only one "white site" that has not been replicated being the urban site not comparable to the MPA due to the high anthropogenic pressure it presents. It does not detract from the value of the study and the statistical design seems to me to be correct except for the fact of considering both unprotected localities as controls. In this sense, the results can only be generalized under a greater bibliographic support in the discussion and a greater comparison of these results with the existing literature.
The procedure for calculating the cover percentage of Cystoseira s.l. species in each square (50 x 50 cm) is not clear. A figure of the experimental design also indicating the distance covered in each transect would facilitate the reader's understanding.
From the calculated parameters, it is not clear whether D% is calculated as a function of cover or number of individuals. The formula used could be represented to facilitate the reader's understanding. If it is by number of individuals, the procedure used for counting should be explained. The reporting parameter of total abundance is defined with an "N". It is preferable to make some modification so as not to confuse it with the "average percentage cover".
Results:
Detailed data are provided on the abundance and diversity of this group of fucals.
The presentation of table 2 can be improved (The dominance variable is not aligned with the other variables. It isn't easy to follow the values of the columns (I do not know if it can be improved in any way).
In point 2.2 of the results I would say “Comparison of Cystoseira s.l. assemblage between MPA and unprotected sites” instead of "controls".
The appreciations on invasive species are relevant despite not having generated any hypothesis in this regard nor any design to evaluate their distribution pattern.
Discussion.
The results provided, of relevant descriptive and quantitative value, can neither confirm nor invalidate the role of the MPA as a reservoir of Cystoseira diversity. Several sites with disparate protection regimes have been studied without sufficient replication to validate or invalidate the hypotheses raised. However, these data could reinforce these hypotheses when supported not only by the empirical information of the present study but also by the bibliography consulted by the authors of the present study, although this bibliography should be discussed in greater depth and the bibliographic work should also be somewhat more extensive. Along with this, those environmental and geomorphological variables pointed out by the authors that may be key to the distribution of Cystoseira forests and be considered in the design of MPA'S (such as the role of slope or bottom type) should also be further worked on with the existing literature data to see more general patterns and reinforce the authors' hypotheses.
Conclusion
Again, I would replace climate change by global change (for the previously mentioned arguments). The appreciation on the distribution of deep Cystoseira species in some studied sites would perhaps enter more in the introduction except for the indication of the need to extend the research to these depths.
The importance of the studied MPA in particular and of the MPA's in general as preservers of the diversity of Cystoseira s. l in the Mediterranean would gain strength with the mentioned greater bibliographic effort and that would support these hypotheses.
Author Response
Dear reviewer,
We appreciate the feedback and comments you provided, which have enhanced the quality of the manuscript. Our responses follow the reviewer's comments.
Sincerely
Francesco Paolo Mancuso
Reviewer #2:
Comment - This work provides a comprehensive qualitative and quantitative description of the diversity and abundance of Cystoseira fucoids in a specific region of Sicily, Italy. It examines locations with varying anthropogenic pressure, including different areas within a Marine Protected Area (MPA) with a variable protection regime. It also includes a "White site" outside the reserve with low alteration and one highly impacted site close to an urban area. From my perspective, this study lacks replicated locations under comparable conditions, making it inappropriate to designate them as "control sites." Consequently, the results can only be generalized with stronger bibliographic support in the discussion. The observed trends also need more detailed discussion and thorough comparison with existing literature. In the current global change scenario, the qualitative and quantitative information provided in this study is relevant and will enable future research on compositional changes (qualitative and quantitative) in these foundational Mediterranean seaweeds. Overall, I believe this work is publishable but needs restructuring based on the assessments provided.
Response - As emphasized in the conclusion, our study establishes a baseline for understanding the population dynamics of Cystoseira s.l., which is crucial for future monitoring and conservation efforts in the face of climate change. Additionally, as noted by the reviewer, it serves as a pertinent bibliographic reference for further studies on compositional changes in these essential habitat-forming seaweeds.
Below, I explain the aspects that I think can be improved in each section:
Abstract:
Comment - It should be restructured with the new contributions from the introduction and discussion. While the abstract emphasizes the role of climate change in driving Cystoseira communities, in the introduction authors recognize various anthropogenic stressors such as overfishing, urbanization, and pollution. Therefore, it might be more appropriate to refer to "global change" rather than "climate change."
Response – Thank you for the observation. We modified climate change with global change.
Introduction:
Comment - The hypothesis that Cystoseira s.l. will be more diverse and structured within the MPA and most protected situations is proposed, but the causal mechanisms supporting this hypothesis need clarification, along with adequate bibliographic support.
Response – We modified the introduction including mechanisms that can support our hypothesis. “Marine Protected Areas (MPAs) have become highly important tools for biodiversity conservation and management worldwide [26]. MPAs typically restrict human activities such as overfishing and urbanization, allowing natural regeneration and maintenance of marine ecosystems, fostering greater diversity [27]. In the Mediterra-nean Sea, a large part of the research concerning the effectiveness of MPAs has focused on fish assemblages [28], while few studies focused on foundation seaweeds [15,29,30]. MPAs can also play a critical role in the conservation of Cystoseira s.l. forests, as they guarantee protection from various anthropogenic impacts such as over-fishing and urbanization [27,29]. The protective environment within MPAs may facilitate higher recruitment and reproduction rates of Cystoseira s.l. populations. Reduced habitat disturbance, in fact, can create conditions conducive to successful re-production and the establishment of new individuals [27]. Moreover, the better protection provided by MPAs compared to the non-managed sites makes them excellent areas where restoration actions of Cystoseira s.l. forests should be prioritized [27]. Healthy Cystoseira s.l. forests in MPAs may also represent an important source of propagules for the restoration of lost or degraded forests outside MPA borders, a crucial aspect due to the low dispersal capacity of most of these species [15,22,31,32].”
Comment - The concept of structure should be defined in the introduction as it is an ambiguous concept.
Response – Thank you for the observation. We removed the term “structured” because do not add important information for the results provided along the text.
Comment - On the other hand, the economic value of Cystoseira communities is discussed to emphasize their importance for conservation, citing a study (25) that draws an analogy between the energy investment required for ecosystem maintenance and a transformation of this investment into monetary units.Despite acknowledging the relevance of the cited study, I do not believe it can be stated in the introduction that these communities have direct economic value, as they are not commodities directly entering the market system and, therefore, cannot possess such value. I think the aspect of monetizing natural communities (which presents many implementation problems, see, for example, Vatn 2000: https://doi.org/10.3197/096327100129342173) is not the focus of this study. I would omit the mention of "economic value" in line 51.
Response – Thank you for your observation. We modified the sentence removing reference about the economic value of Cystoseira s.l. “This shift results in a decrease in essential ecosystem services [14,15].”
Materials and Methods:
Comment -There are 3 zones (ABC) depending on the degree of protection within the MPA. However, in Figure 1 these zones (A, B, C) are not marked. It would be easier to understand if this information were also in Figure 1 (although these zones are well described in the text). On the other hand, more detail on the type of protection regime present in zones B and C is lacking.
Response – Figure 1 show the areas with different levels of protection as reported in the label “Within the MPA, red, yellow, and blue areas denote no-take, general, and limited protection zones, respectively” as well as the sites investigated. However, we modified the label of Figure 1 to be more clear “Figure 1. Study sites (red dots) along the northwester rocky-shore of Sicily: MPA = MPA of “Capo Gallo-Isola delle Femmine”, MC = Monte Cofano, PP = Punta Priola. The top panel displays information about the sites investigated within the MPA: IdF = Isola, LC = La Cala, PM = Punta Matese, PB = Punta Barcarello, B = Barcarello, CG = Capo Gallo. Within the MPA (upper panel), red, yellow, and blue areas denote no-take (zone A), general (zone B), and limited (zone C) protection zones, respectively.”
Comment - The unprotected locations are contrasted from the perspective of anthropogenic impact. It must be said, therefore, that this study does not have replicated localities under comparable conditions, nor can we speak, in my opinion, of "control sites" since there is only one "white site" that has not been replicated being the urban site not comparable to the MPA due to the high anthropogenic pressure it presents. It does not detract from the value of the study and the statistical design seems to me to be correct except for the fact of considering both unprotected localities as controls. In this sense, the results can only be generalized under a greater bibliographic support in the discussion and a greater comparison of these results with the existing literature.
Response – Thank you for your observation. We believe that confusion arises from the use of the word "control." We have decided to remove the word "control" and refer to them as "unprotected" sites.
Comment - The procedure for calculating the cover percentage of Cystoseira s.l. species in each square (50 x 50 cm) is not clear. A figure of the experimental design also indicating the distance covered in each transect would facilitate the reader's understanding.
Response – We added information in the material and methods section: “Cover was estimated divided the quadrat into 25 equal squares: we attributed a cover score from 0 to 4 to each square, and then summed up scores where the taxon was present. Organisms filling< 1⁄4 square were given the value of 0.5 [16].” Moreover, we add Figure S1 in supplementary materials to illustrate sampling design.
Comment - From the calculated parameters, it is not clear whether D% is calculated as a function of cover or number of individuals. The formula used could be represented to facilitate the reader's understanding. If it is by number of individuals, the procedure used for counting should be explained. The reporting parameter of total abundance is defined with an "N". It is preferable to make some modification so as not to confuse it with the "average percentage cover".
Response – Thank you for your observation. Analysis was made on the cover percentage data collected. We clarify it both on the caption of Table 2 adding the sentence “Calculations are based on Cystoseira s.l. cover percentage using 50x50 cm quadrats (n = 4).” and in the Data analysis section “For each investigated depth range, the abundance (N, average percentage cover), frequency (F%, the percentage of samples in which a particular species is present), and dominance index (D%, cover percentage of a particular species to the total cover percentage of Cystoseira s.l. species within the sample) for each Cystoseira s.l. species were estimated [17]. Additionally, for each area investigated (MPA’s zone and control sites), Cystoseira s.l. species were characterized based on abundance (N; expressed as cover percentage), rarefied species richness (S), Shannon-Wiener diversity index (H′), and Pielou’s Evenness index (J).”
Results:
Comment -Detailed data are provided on the abundance and diversity of this group of fucals.
Response – Thanks.
Comment - The presentation of table 2 can be improved (The dominance variable is not aligned with the other variables. It isn't easy to follow the values of the columns (I do not know if it can be improved in any way).
Response – We modified Tables 1 and 2 also following the comments of the other two reviewers. We believe final appearance will be improved by the journal.
Comment - In point 2.2 of the results I would say “Comparison of Cystoseira s.l. assemblage between MPA and unprotected sites” instead of "controls".
Response – thanks. We modified the point 2.2 as “Comparison of Cystoseira s.l. assemblage between MPA and unprotected sites”
Comment - The appreciations on invasive species are relevant despite not having generated any hypothesis in this regard nor any design to evaluate their distribution pattern.
Response – Thank you for your observation. We added the following sentence at the end of the introduction section: “Finally, in the sites investigated, we documented the presence of non-indigenous seaweeds, providing crucial information that offers a clearer understanding of the current challenges.”
Discussion.
Comment - The results provided, of relevant descriptive and quantitative value, can neither confirm nor invalidate the role of the MPA as a reservoir of Cystoseira diversity. Several sites with disparate protection regimes have been studied without sufficient replication to validate or invalidate the hypotheses raised. However, these data could reinforce these hypotheses when supported not only by the empirical information of the present study but also by the bibliography consulted by the authors of the present study, although this bibliography should be discussed in greater depth and the bibliographic work should also be somewhat more extensive. Along with this, those environmental and geomorphological variables pointed out by the authors that may be key to the distribution of Cystoseira forests and be considered in the design of MPA'S (such as the role of slope or bottom type) should also be further worked on with the existing literature data to see more general patterns and reinforce the authors' hypotheses.
Response - Thank you for your observations. We included more details and scientific references to support our results in the discussion section. Following the revised section: “The comparison with unprotected sites revealed differences that were more evident when comparing the MPA sites with the Punta Priola site than with the site of Monte Cofano. The vegetation in Punta Priola was mainly characterized by high sedimentation rates, with Dictyopteris polypodioides being the main macroalgal species covering large part of the rocky substrate, which appears to inhibit the growth of other seaweeds (Figure 9B). This pattern aligns with findings from previous studies, which emphasize the role of sedimentation in shaping not only Cystoseira s.l. populations [18] but macroalgal communities in general. Sedimentation is shown to inhibit zoospore adhesion [19] and facilitate negative interactions by promoting turf-forming algae, which, in turn, inhibit canopy-forming macrophytes [20–22] The absence of E. amentacea, one of the most important Cystoseira s.l. species used to measure water quality, further revealed the site's impacted state [23]. Water quality has been shown to reduce the survival and growth of Cystoseira s.l. [24], which would then justify the low diversity of these habitat-forming seaweeds observed in Punta Priola. Furthermore, the proximity of this site to urban center can facilitate the presence of multiple co-occurring anthropogenic stressors likely drivers the poor Cystoseira s.l. conditions of this site [25]. In contrast, our findings demonstrate that Cystoseira s.l. populations are comparable between the unprotected site of Monte Cofano and the sites within the “Capo Gallo - Isola delle Femmine” MPA (Figure 9A). The presence of healthy and dense Cystoseira s.l. forests in this unprotected site suggests that anthropogenic disturbances, such as trampling, harvesting, pollution, and overgrazing, are relatively limited at Monte Cofano. This finding confirms that robust forests thrive in non-protected, naturally isolated, and lightly disturbed locations [9].”
Conclusion
Comment - Again, I would replace climate change by global change (for the previously mentioned arguments). The appreciation on the distribution of deep Cystoseira species in some studied sites would perhaps enter more in the introduction except for the indication of the need to extend the research to these depths. The importance of the studied MPA in particular and of the MPA's in general as preservers of the diversity of Cystoseira s. l in the Mediterranean would gain strength with the mentioned greater bibliographic effort and that would support these hypotheses.
Response – Thank you for the suggestion. We replaced the term climate change with global change. Moreover, we modified the introduction highlighting the importance of MPAs in preserving Cystoseria s.l. populations

Reviewer 3 Report
Comments and Suggestions for Authors
This is an important paper; as the authors point out, there are few studies of the effectiveness of Marine Protected Areas.
Items needing correction or clarification.
Line 16. Delete first ‘the’ on this line.
Lines 44. A reference is needed for the longevity of Cystoseira s.l., with values for the range of longevities.
Tables 1 and 2. Although it is implicitly defined on line 369, ‘white’ should be defined in these tables.
Table 2. What is meant by ‘N’, ‘F%’ and ‘%’.
Lines 186 and 187. Does ‘aerocysts’ mean ‘gas bladders’?
Line 240. ‘as on’ not ‘the’.
Line 321. Italicize ‘Caulerpa cylindracea’.
Line 342. Clarify ‘high naturalistic interest’.
Line 358. ‘effectiveness’, not ‘efficiency’.
Comments on the Quality of English LanguageSlight improvment needed
Author Response
Dear reviewer,
We appreciate the feedback and comments you provided, which have enhanced the quality of the manuscript. Our responses follow the reviewer's comments.
Sincerely
Francesco Paolo Mancuso
Reviewer #3:
This is an important paper; as the authors point out, there are few studies of the effectiveness of Marine Protected Areas.
Items needing correction or clarification.
Comment - Line 16. Delete first ‘the’ on this line.
Response – Done.
Comment - A reference is needed for the longevity of Cystoseira s.l., with values for the range of longevities.
Response – thank you for the observation. We modified the sentence as follows: “Like other coastal marine ecosystems, these long-lived (between one and more than five decades [26,27]) primary producers…”
Comment - Tables 1 and 2. Although it is implicitly defined on line 369, ‘white’ should be defined in these tables.
Response – Thank you for the observation. We modified the Tables labels as follow: Table 1. List of the recorded Cystoseira s.l. species across MPA zones (A, B, and C) and control sites (white = natural and less impacted site, urban = highly impacted site close to the urban center). Taxonomy and nomenclature were updated according to the AlgaeBase [28] database. Table 2. Average abundance (N), frequency of occurrence (F %) and dominance (D %) of the Cystoseira sl. species identified on MPA zones (A, B, and C) and control sites (white = natural and less impacted site, urban = highly impacted site close to the urban center).
Comment - Table 2. What is meant by ‘N’, ‘F%’ and ‘%’.
Response – The meaning of N, F% and D% was already specified in the caption of Table 2 “Average abundance (N), frequency of occurrence (F %) and dominance (D %) of the Cystoseira sl. species identified on each site.”
Comment - Lines 186 and 187. Does ‘aerocysts’ mean ‘gas bladders’?
Response - To our knowledge "gas bladder" is a term commonly used for fish. We prefer to use the term "aerocyst” because is commonly used with algae where gas-filled structures provide buoyancy.
Comment - Line 321. Italicize ‘Caulerpa cylindracea’.
Response – Done.
Comment - Line 342. Clarify ‘high naturalistic interest’.
Response – We modified the sentence as follow: “Due to the limestone nature, flowing waters generate karst phenomena, leading to caves of significant ecological importance..”
Comment - Line 358. ‘effectiveness’, not ‘efficiency’.
Response – Done.

Round 2
Reviewer 2 Report
Comments and Suggestions for Authors
In this second version of the manuscript, the authors have successfully addressed all those aspects that I think were improvable (the substitution of "control sites" to "unprotected sites" should be checked, however, as it has not been corrected throughout the text).In my opinion, this has given the article a good scientific soundness. In addition, the information provided about this sensitive group of fucoid algae will be useful for local long-term studies but also to reproduce this type of study in other localities increasing the knowledge about this important group of foundation species. I therefore recommend the publication of this work.